# Structural basis of transcription arrest by coliphage HK022 Nun in an *Escherichia coli* RNA polymerase elongation complex

Jin Young Kang[1], Paul Dominic B Olinares[2], James Chen[1], Elizabeth A Campbell[1], Arkady Mustaev[3,4,5], Brian T Chait[2], Max E Gottesman[6], Seth A Darst[1]*

[1]Laboratory of Molecular Biophysics, The Rockefeller University, New York City, United States; [2]Laboratory of Mass Spectrometry and Gaseous Ion Chemistry, The Rockefeller University, New York City, United States; [3]Public Health Research Institute, Newark, United States; [4]Department of Microbiology and Molecular Genetics, Rutgers Biomedical and Health Sciences, Newark, United States; [5]Rutgers New Jersey Medical School, Rutgers Biomedical and Health Sciences, Newark, United States; [6]Department of Microbiology and Immunology, Columbia University Medical Center, New York City, United States

*For correspondence: darst@rockefeller.edu

**Abstract** Coliphage HK022 Nun blocks superinfection by coliphage λ by stalling RNA polymerase (RNAP) translocation specifically on λ DNA. To provide a structural framework to understand how Nun blocks RNAP translocation, we determined structures of *Escherichia coli* RNAP ternary elongation complexes (TECs) with and without Nun by single-particle cryo-electron microscopy. Nun fits tightly into the TEC by taking advantage of gaps between the RNAP and the nucleic acids. The C-terminal segment of Nun interacts with the RNAP $\beta$ and $\beta'$ subunits inside the RNAP active site cleft as well as with nearly every element of the nucleic acid scaffold, essentially crosslinking the RNAP and the nucleic acids to prevent translocation, a mechanism supported by the effects of Nun amino acid substitutions. The nature of Nun interactions inside the RNAP active site cleft suggests that RNAP clamp opening is required for Nun to establish its interactions, explaining why Nun acts on paused TECs.

## Introduction

Bacteriophages are the most common and diverse entities in the biosphere. They represent the simplest systems that are capable of self-replication. Many of the paradigms of transcription regulation were originally discovered studying bacteriophages (*Ptashne, 2005*). Bacteriophage replication inside host bacterial cells involves coopting the cellular transcription machinery. Generally, this is achieved through small, phage-encoded proteins that bind to the host RNA polymerase (RNAP) and alter its properties. Elucidating these processes provides insight into the mechanism and regulation of bacterial transcription itself.

The 13 kDa Nun protein of coliphage HK022 has one known function, to exclude superinfection by the rival coliphage λ (*Robert et al., 1987*) by arresting host *Escherichia coli* (*Eco*) RNAP transcription specifically on λ DNA (*Hung and Gottesman, 1997*; *Vitiello et al., 2014*). HK022 Nun competes with, and antagonizes the function of, the λ N antitermination protein (*Robert et al., 1987*). Like λ N, Nun achieves specificity by binding to the *cis*-acting boxB RNA hairpin within *nut* (N-utilization) RNA sequences in λ nascent transcripts through an N-terminal arginine-rich motif, or ARM (*Chattopadhyay et al., 1995*; *Faber et al., 2001*; *Schärpf et al., 2000*). Unlike λ N, which

suppresses transcription termination at terminators downstream of the *nut* sites, Nun provokes premature downstream termination (*Robert et al., 1987*; *Robledo et al., 1991*).

After localization to specific ternary elongation complexes (TECs) containing the λ *nut* sequence, the C-terminal segment of Nun interacts with the TEC to block RNAP translocation (forward or reverse) at intrinsic pause sites without affecting RNAP catalytic activity and without dissociating the complex (*Hung and Gottesman, 1997*; *Vitiello et al., 2014*). Nun-mediated transcription termination in vivo depends on the function of the Mfd protein, which can remove the arrested TECs from the DNA template (*Washburn et al., 2003*). At normal in vivo expression levels, Nun acts only on TECs containing the λ *nut* sequence, and Nun function is dependent on NusA, NusB, NusE, and NusG (*Robert et al., 1987*; *Robledo et al., 1991*). At high expression levels in vivo, Nun is toxic to *Eco* cells as it acts to arrest transcription elongation by the host RNAP nonspecifically (*Uc-Mass et al., 2008*). Nun alone arrests *Eco* TECs without λ N *nut* RNA elements and independent of Nus factors when present at high concentration in vitro (*Hung and Gottesman, 1997*).

We used single-particle cryo-electron microscopy (cryoEM) to determine structures of *Eco* TECs (4.1 and 4.4 Å resolution) and a Nun/TEC complex (3.7 Å resolution). The structures, combined with the biochemical effects of amino acid substitutions in Nun, explain how Nun blocks both forward and reverse translocation of the RNAP on the nucleic acids and also provides insight into why Nun acts primarily on paused TECs.

## Results

### Nun acts on an *Eco* TEC assembled from a minimized nucleic acid scaffold

To assemble *Eco* TECs suitable for structure determination and sensitive to Nun, we modified a 65-nucleotide (nt) nucleic acid scaffold (na-scaffold) sequence shown previously to respond to Nun in vitro (*Vitiello et al., 2014*) by shortening the DNA template to 29-nt (−15 to +14, where +1 denotes the position corresponding to the 3′-end of the pre-translocated RNA transcript) and engineering a stable, 10-nt transcription bubble (−9 to +1) to accommodate a 9-nt post-translocated RNA transcript by introducing non-complementarity in the DNA non-template strand (nt-strand; *Figure 1A*). The final na-scaffold included a six base pair (bp) upstream DNA duplex (−15 to −10) and a 13 bp downstream DNA duplex (+2 to +14).

*Eco* TECs assembled from purified *Eco* core RNAP and the na-scaffold with a 9-nt post-translocated RNA transcript (TEC-9A, since the RNA is 9-nt in length and contains A at the 3′-end; *Figure 1A* , 9A RNA) were active in transcription elongation (*Figure 1B*, lane 2). Under the same conditions in the presence of a saturating concentration of Nun (20 μM), essentially 100% of the post-translocated 9A transcript was extended to 10C as expected (*Vitiello et al., 2014*), since this reaction did not require RNAP translocation on the template (*Figure 1B*, compare lanes 3 and 4). However, further extension of 10C was efficiently inhibited by Nun (*Figure 1B*, lane 4). We conclude that Nun alone efficiently binds the *Eco* TEC-9A and inhibits RNAP translocation, indicating that our reconstituted system captures Nun function.

### The Nun:TEC stoichiometry is 1:1

Prior crosslinking analyses left open the possibility of a Nun:TEC stoichiometry of 1:1 or 2:1 (*Mustaev et al., 2016*). To address this, we performed native mass spectrometry analysis of the TEC with and without addition of Nun. Two major charge-state series corresponding to experimental masses of 383,971 ± 8 Da and 396,705 ± 6 Da were observed in the mass spectrum of Nun/TEC (*Figure 1—figure supplement 1*). The lower mass (384.0 kDa) matches the expected mass of the fully assembled TEC within the accuracy of the measurement (0.05%). Importantly, the difference between the two measured masses is 12,734 Da, which is close to the expected mass of Nun (12,732 Da), revealing that the 396.7-kDa assembly consists of TEC with one Nun bound.

### Structure determination of the *Eco* TEC ± Nun

We used single-particle cryoEM to determine structures of *Eco* TECs with and without Nun. Preliminary cryoEM analyses led to the following insights: (1) TEC-9A exhibited disorder in the β-flap

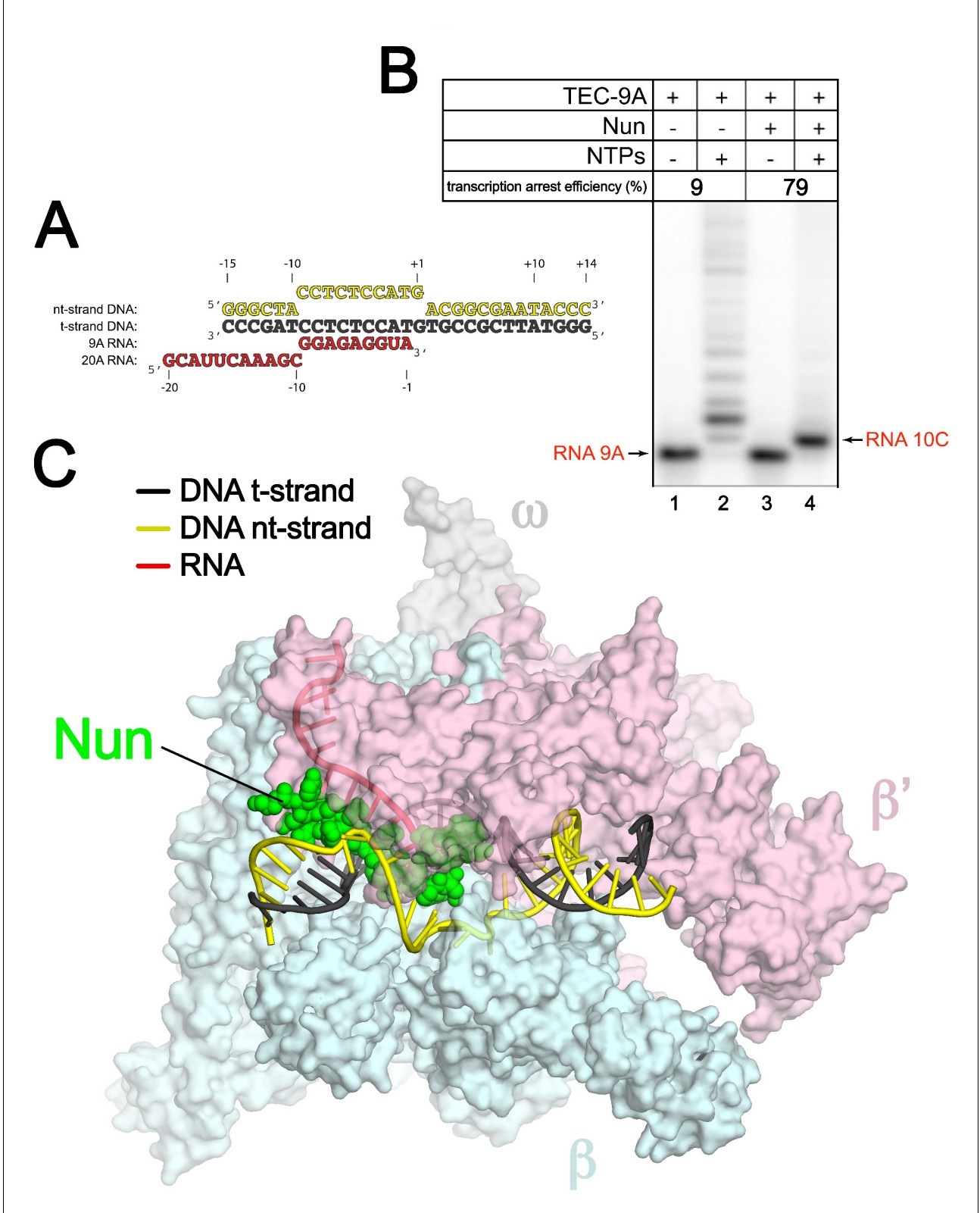

**Figure 1.** An *Eco* TEC with HK022 Nun. (**A**) Na-scaffold – Synthetic oligonucleotides used for *Eco* TEC reconstitution and the numbering scheme used throughout this manuscript. The DNA sequence, a shortened version of the na-scaffold used by (*Vitiello et al., 2014*), is derived from the T7 A1 transcription unit. The nt-strand DNA (colored yellow) contains an engineered non-complementary transcription bubble from −9 to +1. TEC-9A was reconstituted using the 9A RNA (−9 to −1). TEC-20A was reconstituted using the 20A RNA (−20 to −1). (**B**) TEC-9A is sensitive to Nun. [32]P-labeled

*Figure 1 continued on next page*

*Figure 1 continued*

RNAs were monitored by polyacrylamide gel electrophoresis and autoradiography. TEC-9A was assembled using 5′-$^{32}$P-labeled 9A RNA (lane 1). Addition of ATP, CTP, GTP, and UTP (NTPs) resulted in elongation into longer RNAs (lane 2). Nun/TEC-9A was form by the addition of a saturating concentration of Nun (lane 3). In the presence of all four NTPs, 9A RNA was extended to 10C RNA by the addition of C, but further elongation of the RNA was blocked by Nun (lane 4). The % transcription arrest efficiency was calculated as follows. transcription arrest efficiency (%)=100 x $I_{10C}/I_{total}$. where $I_{10C}$ is the integrated intensity of the band for the 10C RNA (quantitated by phosphorimagery). $I_{total}$ is the total integrated intensity over the whole lane. (C) Overall structure of X-Nun/TEC-20A. The color-coding is denoted in the legend (for the na-scaffold) or by the labels (proteins). The na-scaffold is shown in cartoon format. *Eco* core RNAP is shown as a transparent molecular surface. Nun is shown as CPK atoms. The downstream duplex DNA enters the RNAP from the right, the upstream duplex DNA exits the complex to the left.

The following figure supplements are available for figure 1:

**Figure supplement 1.** Native MS analysis of TEC with and without incubation of Nun.

**Figure supplement 2.** CryoEM of X-Nun/TEC-20A.

**Figure supplement 3.** CryoEM of X-TEC-20A.

**Figure supplement 4.** CryoEM of TEC-20A.

**Figure supplement 5.** Comparison of TEC-20A, X-TEC-20A, and X-Nun-TEC-20A structures (also see *Supplementary file 2*).

**Figure supplement 6.** Overall cryoEM map and local resolution of X-Nun/TEC-20A.

**Figure supplement 7.** Overall cryoEM map and local resolution of X-TEC-20A.

domain, possibly due to the absence of upstream RNA filling the RNA exit channel underneath (*Lane and Darst, 2010a*); (2) Density attributable to Nun was weak and fragmented, possibly due to dissociation during sample preparation, resulting in low occupancy. We used 20A RNA (*Figure 1A*) to assemble TEC-20A for cryoEM structure determination to fill the RNA exit channel with the upstream single-stranded RNA (ssRNA, −10 to −20; *Figure 1A*) and to visualize how the RNAP disrupts the RNA:DNA hybrid and directs the upstream ssRNA into the RNA exit channel (*Korzheva et al., 1998*; *Richardson, 1975*). TEC-20A, like TEC-9A, is active in elongation and Nun-mediated arrest (*Figure 1—figure supplement 2A*). We used mild glutaraldehyde crosslinking to stabilize the Nun/TEC complex (*Figure 1—figure supplement 2B*), resulting in high Nun occupancy.

We present three cryoEM reconstructions, glutaraldehyde crosslinked Nun/TEC-20A (X-Nun/TEC-20A; 3.7 Å overall resolution; *Figure 1C*, *Figure 1—figure supplement 2*; *Supplementary file 1*), glutaraldehyde crosslinked TEC-20A (X-TEC-20A; 4.1 Å overall resolution; *Figure 1—figure supplement 3*; *Supplementary file 1*), and TEC-20A without glutaraldehyde crosslinking (4.4 Å overall resolution; *Figure 1—figure supplement 4*; *Supplementary file 1*). Superposition of all three structures by RNAP Cα positions resulted in root mean square deviations of <0.7 Å over at least 2750 Cα's (*Supplementary file 2*). From the structural correspondence between TEC-20A and X-TEC-20A, we conclude that the crosslinked structures are representative of the uncrosslinked solution structures (i.e. the crosslinking did not trap rare conformations; *Figure 1—figure supplement 5A*). Moreover, most, if not all of the Nun molecules were crosslink free (note the presence of apparently stoichiometric non-crosslinked Nun evident in the sample; *Figure 1—figure supplement 2B*), making it unlikely that the crosslinking introduced non-native contacts. From the structural correspondence between TEC-20A, X-TEC-20A, and X-Nun/TEC-20A, we conclude that the TEC accommodates Nun without global conformational changes (*Figure 1—figure supplement 5B*). Localized conformational changes will be discussed below.

The overall resolution of the X-Nun/TEC-20A cryoEM density map was 3.7 Å (*Figure 1—figure supplement 2F*) but local resolution calculations indicate much of the central core of the structure, including the active site, the 9 bp RNA/DNA hybrid, and much of the visible part of Nun, was determined to 3.3–3.6 Å resolution (*Figure 1—figure supplement 6*). The overall resolution of the X-TEC-

20A density map was 4.1 Å (*Figure 1—figure supplement 3F*), but the central core of the structure was determined to 3.6–4.0 Å resolution (*Figure 1—figure supplement 7*).

### *Eco* TEC structure

In TEC-20A, the downstream duplex DNA enters the active site cleft and begins unwinding at +1 (*Figure 2A*). The nt-strand G at +1 binds in a G-specific pocket of the $\beta$-subunit (*Figure 2—figure supplement 1A*) as observed in initiation complex structures (*Bae et al., 2015*; *Zhang et al., 2012*). A G at this position of the na-scaffold facilitates Nun-mediated translocation arrest (*Figure 2—figure supplement 1B*). The t-strand passes over the bridge helix (BH) to position the +1 t-strand base in the RNAP active site. The 9-bp RNA/DNA hybrid is in the post-translocated state, with the 3'-end of the RNA at −1 (*Figure 2A*, *Figure 2—figure supplement 2*).

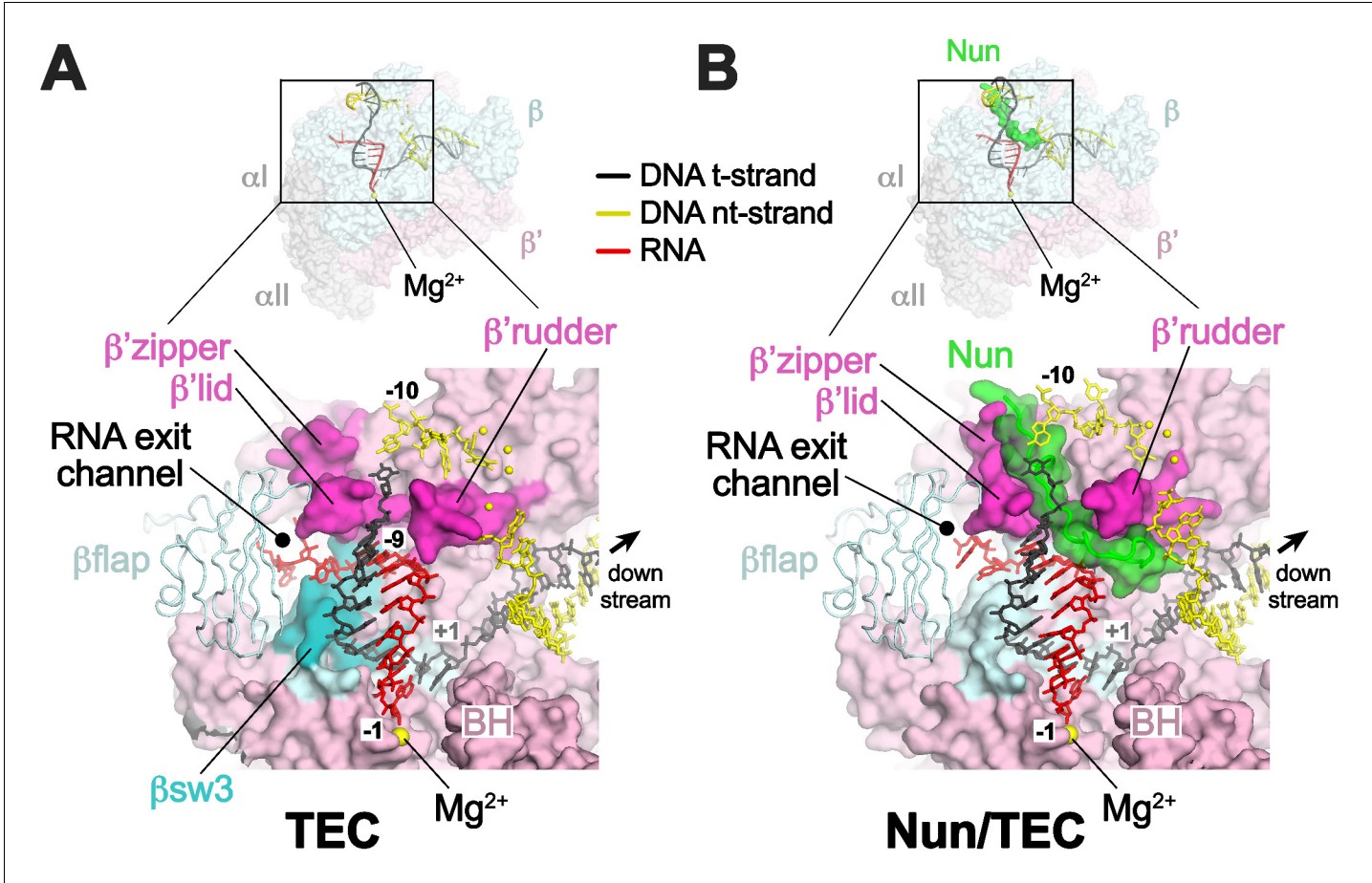

**Figure 2.** Comparison of X-TEC-20A and X-Nun/TEC-20A structures. (A) (top) Overall view of the X-TEC-20A structure. The RNAP is shown as a transparent molecular surface, revealing the na-scaffold inside (shown in cartoon format). The boxed region is magnified below. (bottom) Magnified view of boxed region from overall view above. Most of the $\beta$ subunit has been removed to reveal the inside of the RNAP active site cleft. Most of the RNAP is shown as a molecular surface, with $\beta$ colored light cyan but with switch3 ($\beta$sw3) colored light blue, $\beta'$ colored light pink but with the $\beta'$zipper, $\beta'$lid, and $\beta'$rudder colored light magenta. The Bridge-Helix (BH) is also labeled. The $\beta$flap, which covers the RNA exit channel, is shown as a backbone worm. The na-scaffold is shown in stick format, color-coded as shown in the legend. The RNAP active-site $Mg^{2+}$-ion is shown as a yellow sphere. (B) (top) Overall view of the X-Nun/TEC-20A structure. Nun is shown as a green molecular surface (bottom). Same as A (bottom) except Nun is shown as a transparent green molecular surface with an $\alpha$-carbon backbone worm.

The following figure supplements are available for figure 2:

**Figure supplement 1.** The nt-strand ssDNA and Nun.

**Figure supplement 2.** CryoEM density map at the active site.

At the upstream edge of the RNA/DNA hybrid, conserved structural features of the RNAP limit the extent of the hybrid and provide tracks to confine and direct the ssRNA and t-strand DNA away from each other (*Figures 2A* and *3*). The β'lid directly blocks the upstream extension of the RNA/DNA hybrid (*Bernecky et al., 2016*; *Vassylyev et al., 2007*; *Westover et al., 2004*). Conserved hydrophobic residues V253 and L255 (*Figure 3—figure supplement 1*) interact with the face of the last upstream RNA/DNA bp at −9, possibly providing a 'greasy' surface for the bases to slip past during translocation. V253 appears to help track the upstream ssRNA towards the RNA exit channel

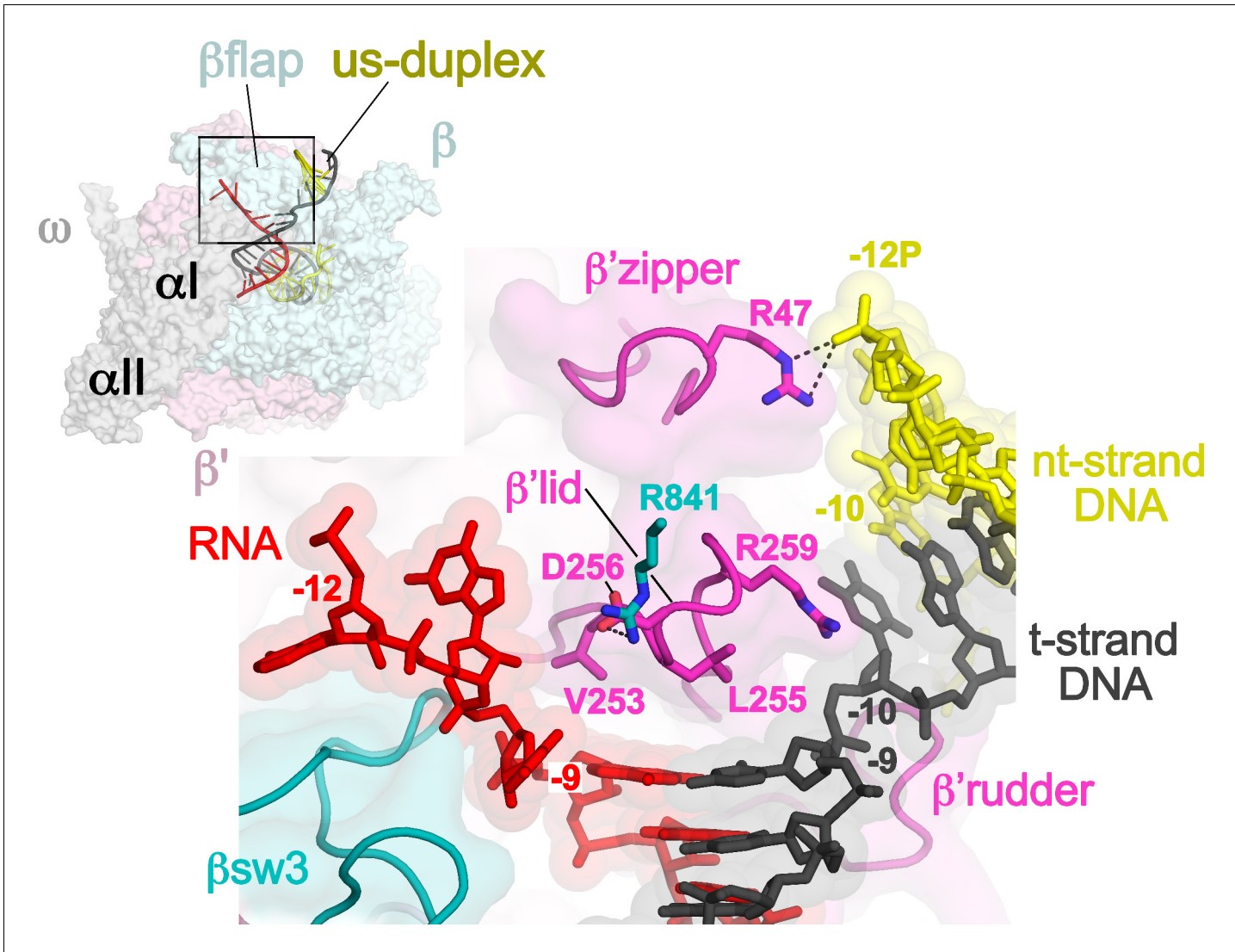

**Figure 3.** Disruption of the RNA/DNA hybrid and formation of the upstream duplex DNA in the TEC. (top left) Overall view of the X-TEC-20A structure. The RNAP is shown as a transparent molecular surface, revealing the na-scaffold inside (shown in cartoon format). The boxed region is magnified below. (bottom right) Conserved structural elements of the RNAP (βsw3, light blue; β'zipper, β'lid, β'rudder, light magenta) disrupt the RNA/DNA hybrid, direct the upstream ssRNA into the RNA exit channel, and direct the upstream t-strand DNA to form the upstream duplex DNA.

The following figure supplements are available for figure 3:

**Figure supplement 1.** Sequence logos showing sequence conservation in RNAP structural elements.

**Figure supplement 2.** Upstream duplex DNA in the TEC and Nun/TEC.

**Figure supplement 3.** TEC model with extended upstream and downstream duplex DNA.

(to the left of the β'lid in *Figure 3*). The RNA track is buttressed on the opposite side by the conserved βswitch 3 (*Figures 2A* and *3*; [*Lane and Darst, 2010a*]). Conserved R259 of theβ'lid (*Figure 3—figure supplement 1*) directs the t-strand DNA away from the RNA (to the right of the β'lid in *Figure 3*), which is buttressed on the opposite side by the β'rudder (*Figures 2A* and *3*). Conserved D256 (*Figure 3—figure supplement 1*), at the top of the β'lid, forms a salt bridge with conserved R841 descending from the βflap, which covers the top of the RNA exit channel (*Figures 2A* and *3*). This and other interactions between the βflap and elements of the β' subunit seal off the nucleic acid tracks, preventing the upstream ssRNA and t-strand DNA from slipping over the top of the β'lid and inappropriately displacing the nt-strand DNA to form an extended RNA/DNA hybrid (*Naryshkina et al., 2006*; *Toulokhonov and Landick, 2006*).

The -10 t-strand nucleotide base pairs with the nt-strand DNA to begin the upstream DNA duplex with no single-stranded gap in the t-strand DNA (*Figure 3*). Density for the upstream DNA duplex is weak and noisy, indicative of mobility. The density maps and refinement results indicate that the -10 bp is distorted (*Figure 3—figure supplement 2A*), consistent with the conclusions of (*Turtola and Belogurov, 2016*). Upstream of the β'lid, conserved R47 of the β'zipper (*Figure 3—figure supplement 1*) interacts with the nt-strand -12 backbone phosphate (*Figure 3*), helping to set the path of the upstream DNA duplex, which exits the complex without any steric clashes, subtending an angle of about 110° with the entering downstream duplex DNA (*Figure 3—figure supplement 3*).

## Nun interacts with RNAP, DNA, and RNA

Although the X-Nun/TEC-20A complex contained full-length Nun (*Figure 1—figure supplement 1*), only the C-terminal 23 residues (residues 87–109), which contain all known determinants for the translocation arrest activity of Nun (*Henthorn and Friedman, 1996*; *Kim and Gottesman, 2004*; *Watnick et al., 2000*), were visible in the structure (*Figure 4A*). Nun forms an extended structure that worms its way into the TEC active site cleft, interacting with the β'zipper, then squeezing through the nt-strand DNA track between the β'lid and the β'rudder, then between the β'rudder and the RNA/DNA hybrid. The very C-terminal segment of Nun (residues 102–109) binds in a pre-existing pocket next to the β'rudder between the downstream duplex DNA and the RNA/DNA hybrid (*Figure 2B*).

The overall structure of the TEC is largely unchanged (*Supplementary file 2*, *Figure 1—figure supplement 5B*), but Nun expands the narrow space between the β'lid, the β'rudder, and the beginning of the upstream duplex DNA by pushing the upstream DNA duplex outwards (away from the RNAP) about 3.5 Å, a full bp step (*Figure 3—figure supplement 2*). Nun residue H93 stacks on the exposed downstream face of the nt-strand DNA -10 base while R94 reaches to the major groove edge of the -10 bp, where it forms a hydrogen-bond (H-bond) with the O4 atom of the t-strand -10T (*Figure 4B*). This interaction is not likely to be sequence-specific since Nun R94 has the reach and flexibility to form a H-bond with a H-acceptor on the major groove edge of either the nt-strand or t-strand base. The presence of Nun in the vicinity of the upstream DNA duplex accounts for the Nun-dependent upstream toe-print (*Vitiello et al., 2014*).

Nun then squeezes between the β'rudder and the RNA/DNA hybrid (*Figure 4C*), displacing some α-carbons of the rudder about 3 Å and altering some rudder/nucleic acid interactions. In particular, in the TEC, β'R322 of the rudder makes polar interactions with the -6 and -7 RNA nucleotides, but in the Nun/TEC complex, β'R322 is displaced and forms an Arg-His pair with Nun H98 (*Heyda et al., 2010*). Nun H98 in turn forms H-bonds with the 2'-OH of the -7 and -8 RNA nucleotides (*Figure 4C*). We tested the validity of this observation by replacing the RNA -7A with deoxy-A (so disrupting the Nun H98/-7 RNA 2'-OH interaction), which did not affect elongation in the absence of Nun in the C10 extension assay, but significantly reduced the transcription arrest efficiency of Nun (*Figure 4—figure supplement 1*). The new position of β'R322 is stabilized by a salt bridge with β'D264 at the base of the lid. This configuration is important since substitutions β'R322H or β'D264G lead to Nun resistance in vivo (*Robledo et al., 1991*). Along with H98, nearby Nun residues N97 and H100 complete a network of polar interactions with the RNA, the t-strand DNA, and the β'rudder (*Figure 4C*).

The negatively-charged C-terminal carboxylate of Nun (S109) forms a salt bridge with Nun R102, resulting in an 8-residue non-covalent circle (*Figure 4A*). The substitution Nun R102A significantly reduced termination efficiency (*Kim and Gottesman, 2004*). Nun residues from N101 to the

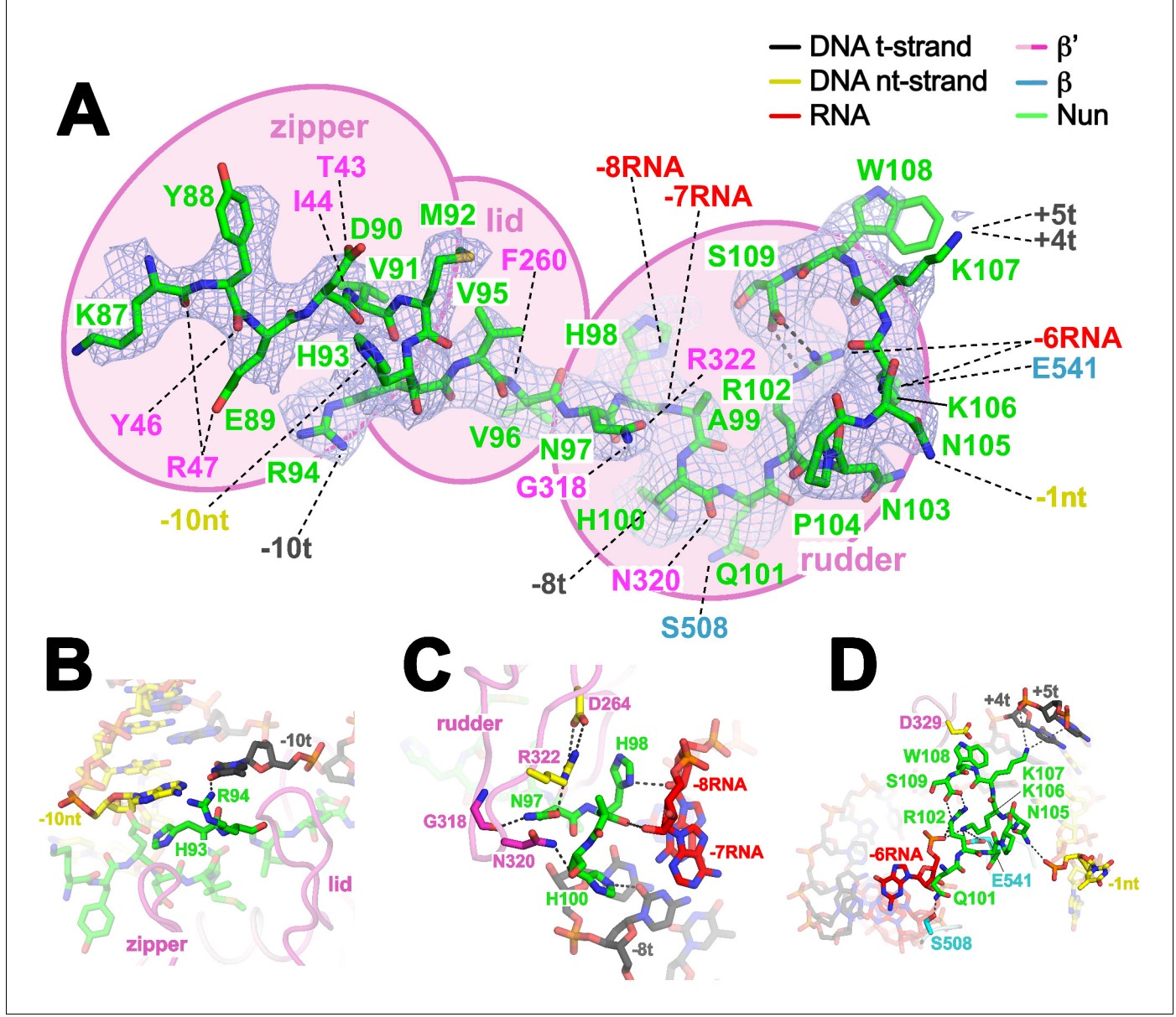

**Figure 4.** Nun cryoEM density and Nun/TEC interactions. (A) The cryoEM density for Nun is shown (blue mesh) with the Nun molecular model (green) superimposed. Nun makes extensive interactions with the $\beta'$ zipper, lid, and rudder of the RNAP (extent of interactions shown schematically by pink ovals). Specific polar interactions (H-bonds and salt bridges) with RNAP residues and with the na-scaffold are denoted (dashed black lines), including the intramolecular Nun interaction between the Nun C-terminus (S109) and Nun R102. (B) Close-up view of interactions between Nun H93 and R94 with the upstream DNA duplex -10 bp. (C) Close-up view of interactions between Nun N97, H98, and H100 with residues of the $\beta'$ rudder, the RNA, and the t-strand DNA. Amino acid substitutions of the $\beta'$ residues colored yellow (R322H or D264G) yield Nun-resistant RNAP *in vivo* (*Robledo et al., 1991*). (D) Close-up view of interactions between Nun Q101-S109 with residues of the RNAP $\beta'$ subunit, the RNA, the nt-strand ssDNA, and the downstream duplex DNA. Amino acid substitutions of the $\beta'$ residues colored yellow (D329G) yields Nun-resistant RNAP in vivo (*Robledo et al., 1991*).

The following figure supplement is available for figure 4:

**Figure supplement 1.** C10 extension assay showing -7 dA in RNA transcript reduces Nun function.

C-terminus form a network of polar interactions with the $\beta$-subunit, the RNA (-6), the downstream duplex DNA (+4t, +5t), and the nt-strand ssDNA (-1nt) (*Figure 4D*). The latter interaction (*Figure 2— figure supplement 1A*) explains why a two nucleotide nt-strand non-complementary 5'-overhang at the downstream fork of the transcription bubble is necessary for optimal Nun function (*Figure 2—*

*figure supplement 1C*). In addition to RNAP substitutions $\beta'$R322H and D264G, the substitution $\beta'$D329G leads to Nun resistance (*Robledo et al., 1991*).$\beta'$D329 forms part of a pocket in the RNAP structure that accommodates Nun W108 (*Figure 4D*). The structure is consistent with previous findings that the C-terminus of Nun is within 15–20 Å of the downstream duplex DNA (*Watnick and Gottesman, 1999*). The location of the Nun C-terminus within the vicinity of the downstream duplex DNA, combined with the finding that normal Nun activity required an aromatic residue at Nun position 108 led to the proposal that Nun-mediated translocation arrest was facilitated by the intercalation of Nun W108 into the downstream DNA (*Watnick and Gottesman, 1999*) but this is not observed in the structure.

A segment of Nun encompassing residues 44 to the C-terminus (residue 109) was found to crosslink to both the 5'- and 3'-ends of RNA transcripts (*Mustaev et al., 2016*). Nun 44-109 crosslinks to the RNA 5'-end when the crosslinking moiety is placed at the -9 to -14 positions. These positions would not be able to crosslink to the Nun C-terminus (residues 87-109) visible in our structure, but Nun residues 44–86 exit the TEC near the RNA exit channel (*Figure 1C*) and could easily reach the upstream RNA, assuming some breathing/opening of the RNA exit channel.

Nun crosslinks to the RNA 3-'end in a 2-nucleotide backtracked complex (*Mustaev et al., 2016*). Structures of both bacterial and eukaryotic TECs backtracked by 1-nucleotide place the backtracked nucleotide (+2 position) in the so-called 'P'-site (*Sekine et al., 2015*; *Wang et al., 2009*). A eukaryotic 2-nucleotide backtracked complex shows the first backtracked nucleotide (+2) again in the P-site, while the second (3') backtracked nucleotide (+3) is disordered, consistent with molecular dynamics simulations showing the +3 nucleotide sampling a large range of conformations (*Wang et al., 2009*). Thus, although the closest atom of Nun to the RNAP active site $Mg^{2+}$ is 27 Å away, the flexible nature of the 2-nucleotide backtracked complex would allow the crosslinking moiety on the base of the RNA 3'-nucleotide to reach Nun.

## Nun mutants retain significant translocation arrest activity

Based on our structural analyses, we generated single (and one double) amino acid substitutions in Nun and tested their activity in vitro. We substituted residues that appeared to make important interactions, including H93A and R94A (*Figure 4B*), N97D, H98A, and H100A (*Figure 4C*), and R102A, R102K, K106A, K107A, K106A/K107A, and W108A (*Figure 4D*). Nun N103 does not appear to make important interactions in the structure (*Figure 4A*), so the N103A substitution served as a control. The Nun substitutions that we tested here included all of the Nun mutants that have been identified as abrogating Nun activity in vivo (*Kim and Gottesman, 2004*; *Watnick and Gottesman, 1999*; *Watnick et al., 2000*). We determined the 'fractional Nun activity' for each mutant (at 20 μM Nun) with the in vitro C10 extension assay (*Figure 1B*) and normalized the results; 0 Nun activity corresponds to no Nun, 1.0 corresponds to the activity of wt-Nun at a saturating concentration (20 μM; *Figure 5A*). The Nun mutants displayed a wide range of activities (*Figure 5—figure supplement 1*), from 21 ± 7(SEM)% for H98A to 96 ± 4% for N103A (essentially wild-type, consistent with our structural analysis).

For the functional mechanism of a transcription factor such as Nun, we can consider two extremes: (1) Binding of the factor to RNAP and the activity of the factor in modulating RNAP function may be inseparable, or (2) binding and activity may be uncoupled. Many examples of the latter case exist; for example, single amino acid substitutions of GreA or GreB severely abrogate their activity in stimulating endonucleolytic transcript cleavage but do not have a noticeable effect on RNAP binding (*Laptenko et al., 2003*; *Opalka et al., 2003*; *Rutherford et al., 2007*; *Sosunova et al., 2003*). To assess the differential effects of the Nun amino acid substitutions on RNAP binding vs. activity in stalling translocation, we used the C10 extension assay to measure the effect of the Nun mutants as a function of their concentration. The resulting isotherms (*Figure 5A*) yielded the effective dissociation constant ($K_d$) and the fractional Nun activity at saturation (normalized with respect to wt-Nun; *Figure 5B*). The effective $K_d$ reveals the effect of the particular Nun substitution on RNAP binding, while the maximum activity reflects the effect of the substitution on Nun activity.

The binding and activity of the Nun mutants are not strictly correlated (*Figure 5B*), indicating that some substitutions have a strong effect on binding but not on activity (R94A), while others affect binding to a lesser degree but have a stronger effect on activity (H98A). Nevertheless,

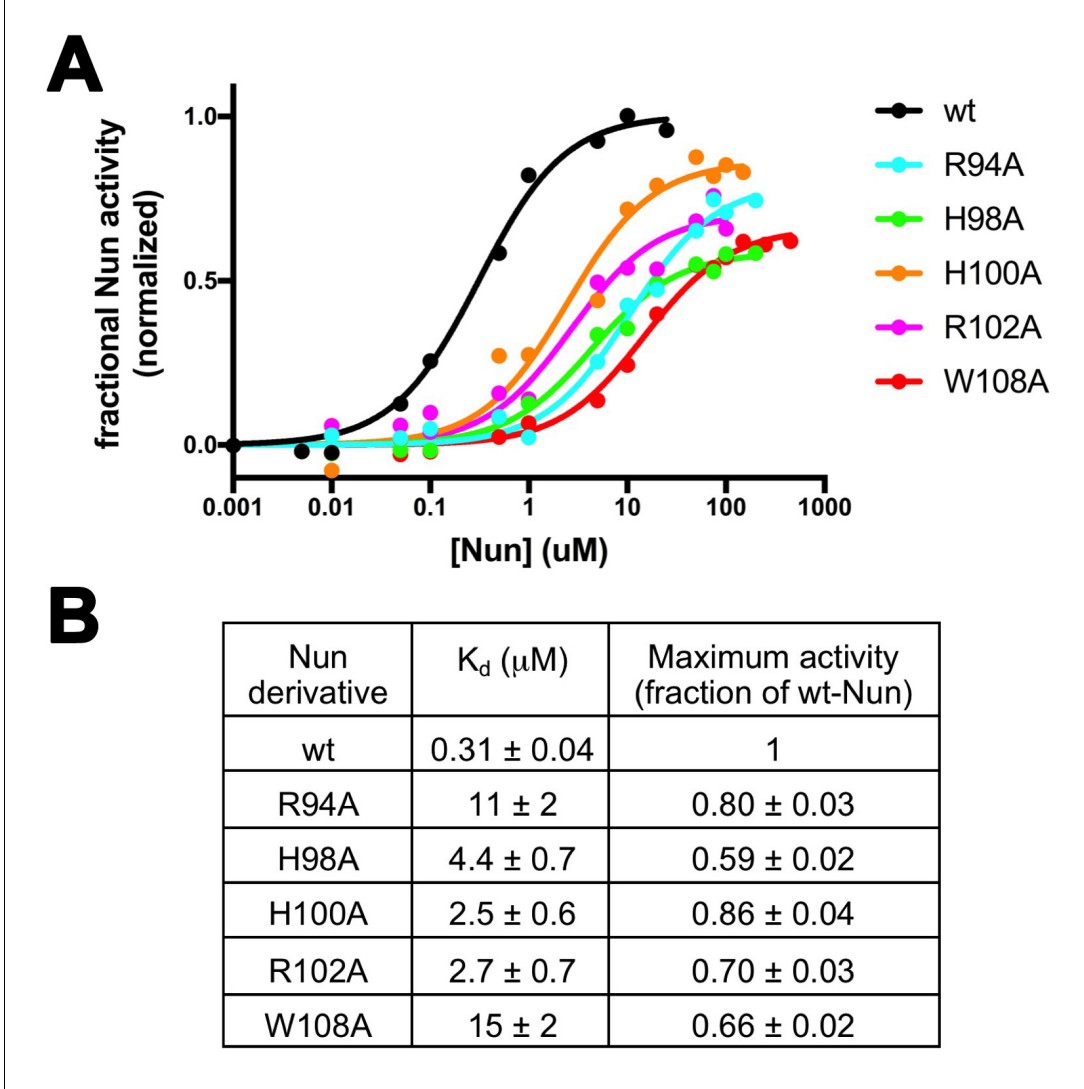

**Figure 5.** Effects of Nun substitutions on TEC binding and stalling of translocation. (**A**) Semi-log plot showing fractional normalized Nun activity vs. Nun concentration for wt-Nun or Nun substitutions (color-coded as shown in the legend). (**B**) Table listing the apparent dissociation constant ($K_d$) and maximum activity (as a fraction of wt-Nun maximum activity) determined from fits of the titration curves shown in (**A**).

The following figure supplement is available for figure 5:

**Figure supplement 1.** Fractional Nun activity for wt-Nun and mutants assayed at 20 µM Nun.

each of the tested mutants show significant translocation arrest activity (59% to 80% of wt-Nun) when assayed at saturating concentration despite widely varying effects on RNAP binding (*Figure 5B*).

## Discussion

### Nun takes advantage of TEC structural requirements dictated by translocation

Efficient transcription elongation requires that the na-scaffold slide freely with respect to the RNAP at each step of translocation in a nucleic acid sequence-independent manner. As a consequence, the RNAP does not form tight, cognate interactions with most of the na-scaffold in the TEC (*Figure 6*).

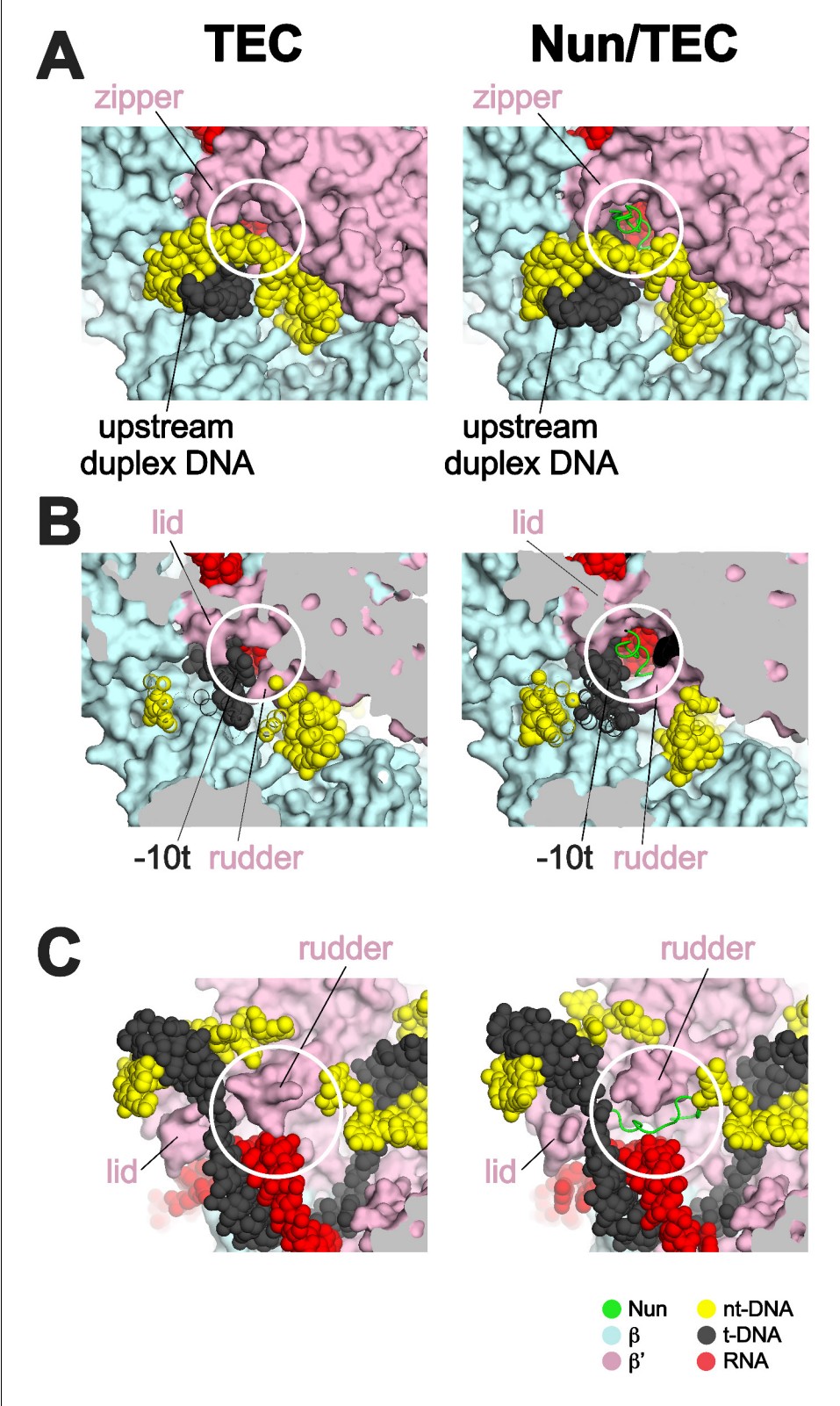

**Figure 6.** Nun takes advantage of gaps between the RNAP and the nucleic acids in the TEC. Views of the X-TEC-20A (left column) and X-Nun/TEC-20A (right column) structures. The RNAP is shown as a molecular surface (color coding shown in the legend below). Nucleic acids are shown in CPK format. Nun is shown as a backbone worm. (**A**) Nun enters the complex through a gap between the zipper and the upstream duplex DNA. (**B**) Nun takes

*Figure 6 continued on next page*

*Figure 6 continued*
advantage of a gap between the lid, the rudder, and the t-strand DNA. (C) The C-terminal part of Nun binds in a pre-existing pocket next to the RNA/DNA hybrid and the rudder.

Nun establishes its interactions with the TEC by taking advantage of the loose RNAP/na-scaffold interactions, insinuating itself into the TEC in pre-existing gaps between the RNAP and the upstream duplex DNA (*Figure 6A*), the t-strand DNA connecting the upstream duplex DNA with the RNA/DNA hybrid (*Figure 6B*), and in a pocket next to the rudder between the RNA/DNA hybrid and the downstream duplex DNA (*Figure 6C*). The Nun/TEC interactions induce some small, localized conformational changes (such as in the upstream duplex, *Figure 3—figure supplement 2*, or in the rudder, *Figures 2* and *6C*), but do not cause large-scale changes in the TEC (*Supplementary file 2*; *Figure 1—figure supplement 5B*).

## Mechanism of Nun translocation arrest

Our results reveal that the functionally critical C-terminal segment of Nun binds to the TEC in an extended conformation, interacting with conserved, functionally important elements of the RNAP ($\beta'$zipper, $\beta'$lid, $\beta'$rudder) as well as with the $\beta$ subunit (*Figure 4*). At the same time, Nun interacts with nearly every element of the na-scaffold: the upstream duplex DNA, the RNA/DNA hybrid, the single-stranded nt-strand DNA within the transcription bubble, and the downstream duplex DNA (*Figures 4* and *7*). With this network of Nun/RNAP and Nun/na-scaffold interactions, Nun essentially mediates extensive crosslinking between the RNAP and the na-scaffold, preventing sliding of the na-scaffold with respect to the RNAP (i.e. preventing translocation). Note that this mechanism explains how Nun arrests both forward and reverse translocation (*Vitiello et al., 2014*). The finding that Nun establishes multiple interactions with both the RNAP and na-scaffold explains how single amino acid substitutions in Nun still retain significant translocation arrest function (*Figure 5*) since the disruption of one or a few interactions still leaves many other interactions that the TEC must overcome in order to translocate (*Figure 7*). Nun does not contain a functional center of critical amino acids for its translocation arrest activity; all of the Nun/TEC interactions contribute to its function.

In phage HK022 lysogens, the concentration of Nun ([Nun]) in the *Eco* cell is likely to be only ~100 nM (*King et al., 2000*). The $K_d$ of wt-Nun for a non-specific TEC (*i.e.* not containing $\lambda$ *nut* boxB RNA) is ~300 nM (*Figure 5*), while the $K_d$ for a TEC containing a boxB RNA hairpin would be expected to be lower. Nun function is stimulated in vivo and in vitro by NusA, NusB, NusE, and NusG, but only in the presence of $\lambda$ *nut* boxA RNA (*Hung and Gottesman, 1995*). The mechanism of Nus factor stimulation of Nun activity is unknown, but the Nus factors likely stabilize Nun binding to $\lambda$ *nut* TECs. These observations indicate that [Nun] in phage HK022 lysogens is precisely tuned to allow Nun to target $\lambda$ *nut* TECs but not cause general translocation arrest throughout the *Eco* genome (*Figure 7B*).

Nun mutants H98A, K106A/K107A, and W108A have been shown previously to be significantly (K106A/K107A) or totally (H98A, W108A) defective in Nun function both in vivo and in vitro (*Kim and Gottesman, 2004*; *Watnick and Gottesman, 1999*; *Watnick et al., 2000*). These results appear to contradict our results that, at saturating [Nun], these mutants exhibit significant translocation arrest activity (H98A, 59%; W108A, 66%; K106A/K107A, >70%; *Figure 5*, *Figure 5—figure supplement 1*). However, in previous studies, In vivo function was assessed using a $\lambda$ *pL-nutL-lacZ* chromosomal transcriptional fusion (*Wilson et al., 1997*) with very low expression levels of the Nun mutants. In vitro transcription reactions included between 50–200 nM Nun mutants. In our study, significant activity of the Nun mutants was observed at saturating [Nun] >250 µM (*Figure 5*). Thus, the previous studies assessed mutant Nun activity at [Nun] too low to overcome the binding defects introduced by the mutations.

## Nun and transcriptional pausing

In the final state of the Nun/TEC complex determined here (*Figure 1C*), Nun forms extensive interactions with both the RNAP clamp domain, the na-scaffold, as well as with elements of the $\beta$ subunit

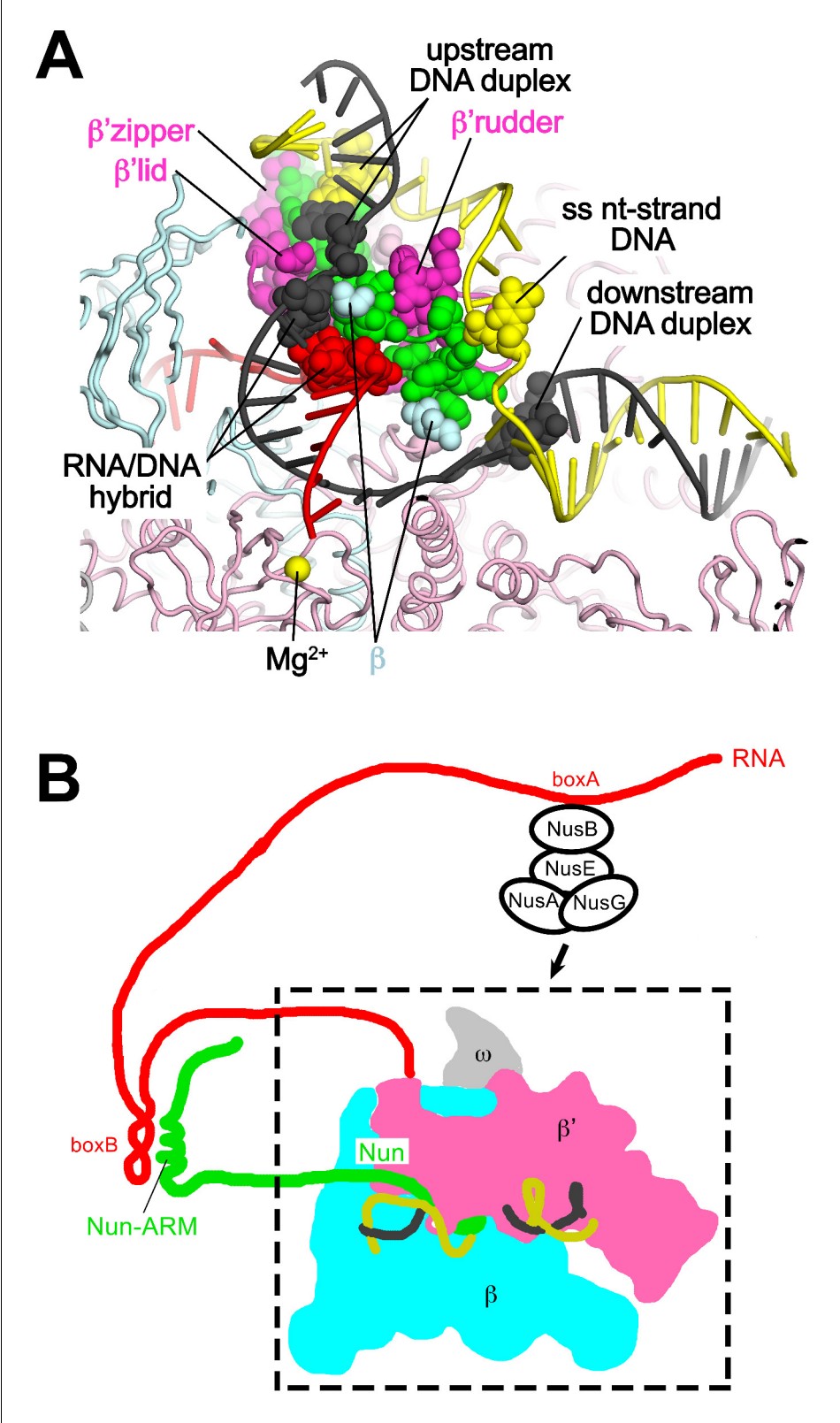

**Figure 7.** Nun mediates extensive interactions between the RNAP and the na-scaffold. (**A**) View inside the RNAP active-site cleft where most of the β subunit has been removed (similar to *Figure 2*). The RNAP (β, light cyan; β', light pink except the β'zipper, β'lid, and β'rudder are light magenta), Nun (green), and the na-scaffold (ntDNA, yellow; tDNA, dark grey; RNA, red) are shown in cartoon format except residues or nucleotides that interact with Nun are shown as CPK atoms. The RNAP active-site $Mg^{2+}$-ion is shown as a yellow sphere. (**B**) Schematic illustrating in vivo action of Nun in the context

**Figure 7 continued**

of a λ *nut* TEC (carrying boxA and boxB RNA sequences) and Nus factors. The X-Nun/TEC structure determined here is contained within the dashed box (RNAP core, DNA template, Nun C-terminal segment). The N-terminal segment of Nun extends from the TEC, where the α-helical Nun-ARM interacts with the λ *nut* boxB RNA hairpin (*Chattopadhyay et al., 1995*; *Faber et al., 2001*), localizing Nun specifically to λ transcripts. Further upstream, the λ *nut* boxA RNA sequence interacts with NusB/E, and together with NusA/G stabilize the Nun/TEC complex and stimulate translocation arrest (*Hung and Gottesman, 1995*).

apposing the clamp (*Figure 7*), tightening and immobilizing normally mobile elements of the TEC, suggesting why the X-Nun/TEC structure was determined to higher resolution than either the X-TEC or the TEC structures (*Figure 1—figure supplements 2–4*). Nun fits tightly into the TEC without causing major conformational changes in the TEC by taking advantage of pre-existing gaps between the RNAP and the na-scaffold (*Figure 6*), but examination of the structure suggests that it would not be possible for Nun to establish itself in this binding mode with the clamp domain tightly closed; clamp opening appears to be required for Nun binding.

There is no clear consensus sequence for Nun-mediated translocation arrest, consistent with the Nun/TEC structure. Nevertheless, Nun does not arrest translocation at random sites on the DNA; Nun preferentially stalls translocation at pre-existing intrinsic transcription pause sites (*Hung and Gottesman, 1995*). Structural (*Weixlbaumer et al., 2013*) and functional (*Hein et al., 2014*) analysis of paused transcription complexes suggests that pausing is linked with a propensity for clamp opening. This provides an explanation for the action of Nun at intrinsic pause sites, since clamp opening is likely required for Nun entry. In addition to clamp opening, TEC pausing provides Nun with a sufficiently long-lived stationary target to establish its extensive network of interactions, which is likely a slow process (*Hung and Gottesman, 1995*).

## Materials and methods

### Protein expression and purification

*Eco* RNAP lacking the αCTDs was prepared as described previously (*Twist et al., 2011*). Glycerol was added to the purified RNAP to 15% (v/v), and the sample was aliquoted and flash-frozen in liquid nitrogen. The aliquots were stored at -80°C until use.

Full-length HK022 Nun was cloned into the pET21d vector (Novagen/EMD Millipore, Billerica, MA) without any tag, and transformed into the *Eco* BL21-AI (arabinose inducible) strain. HK022 Nun was expressed by autoinduction system due to its high toxicity to cells (*Studier, 2005*). The transformed cells were inoculated into non-inducing media (MDAG-135 containing 100 μg/ml ampicillin) and grown overnight. The overnight culture was added to inducing media (ZYM-5052 + 0.05% arabinose+100 μg/ml ampicillin) at a ratio of 1:500 and grown for 16 hr at 37°C. The harvested cells were resuspended in 50 mM Na-phosphate, pH 7.9, 200 mM NaCl, 1 mM DTT, Complete Protease Inhibitor (Roche, Branford, CT) buffer, and lysed in a continuous flow French Press (Avestin, Ottawa, Ontario, Canada). The lysate was centrifuged and the supernatant was loaded onto a HiTrapSP column (GE Healthcare Life Sciences, Pittsburgh, PA). The protein was washed with 50 mM Na-phosphate, pH 7.9, 200 mM NaCl, 1 mM DTT, and eluted with a NaCl gradient to 1 M. The eluted protein was diluted with 50 mM Na-phosphate, pH 7.9, to lower the NaCl concentration to 0.1 M and then loaded onto a MonoS column (GE Healthcare Life Sciences). The protein was eluted with a salt gradient, and the eluted protein was purified by gel filtration chromatography on a HiLoad Superdex75 column (GE Healthcare Life Sciences) with TGE buffer [10 mM Tris-HCl, pH 8, 0.1 mM EDTA, 5% (v/v) glycerol] containing 500 mM NaCl and 1 mM DTT. Nun single or double mutants were generated by overlap or inverse PCR and purified using the same procedure as wild-type Nun. Each Nun mutant was confirmed by mass spectrometry. In the case of Nun K106A/K107A, mass spectrometry showed that the initial sample was >80% T-Nun (Nun with 13 C-terminal amino acids proteolytically removed; (*Watnick and Gottesman, 1998*). T-Nun elutes earlier on the MonoS column, so fractions from the tail-end of the Nun-K106A/K107A MonoS elution peak were analyzed by mass spectrometry and found to be ~50% T-Nun. T-Nun is totally inactive and does not bind RNAP (*Watnick and Gottesman, 1998*), so does not compete with Nun binding, so for the Nun-K106A/

K107A assays (*Figure 5—figure supplement 1*) we adjusted the concentration to compensate. The finding that Nun-K106A/K107A is highly susceptible to in vivo protease cleavage to yield inactive T-Nun may explain the previous result that Nun-K106A/K107A was inactive in vivo (*Kim and Gottesman, 2004*).

### Nucleic acid preparation

Synthetic DNA oligonucleotides were obtained from Integrated DNA Technologies (Coralville, IA), RNA oligonucleotides from GE Healthcare Dharmacon (Lafayette, CO). The nucleic acids were dissolved in RNase-free water (Ambion/ThermoFisher Scientific, Waltham, MA) at 1 mM concentration. To assemble the TEC, template DNA and RNA were mixed at 1:1 ratio, annealed by incubating at 95°C for 2 min, 75°C for 2 min, 45°C for 5 min, and then decreasing the temperature by 2°C for 2 min until reaching 25°C. The annealed template DNA:RNA hybrid was stored at -20°C before use.

### TEC preparation for CryoEM

Purified *Eco* RNAP was buffer-exchanged over the Superose 6 INCREASE (GE Healthcare Life Sciences) column into 20 mM Tris-HCl, pH 8.0, 150 mM KCl, 5 mM $MgCl_2$, 5 mM DTT. The eluted protein was mixed with template DNA:RNA hybrid at a molar raio of 1:1.3 and incubated for 15 min at room temperature. Non-template DNA was added and incubated for 10 min. The complex was concentrated by centrifugal filtration (VivaProducts, Littleton, MA) to 3 mg/ml RNAP concentration before grid preparation.

### Glutaraldehyde Cross-linked TEC preparation for CryoEM

Purified *Eco* RNAP was buffer-exchanged over the Superose 6 INCREASE (GE Healthcare Life Sciences) column into 20 mM HEPES, pH 7.5, 150 mM KCl, 5 mM $MgCl_2$, 5 mM DTT. The eluted protein was mixed with template DNA:RNA hybrid at a molar raio of 1:1.3 and incubated for 15 min at room temperature. Non-template DNA was added and incubated for 10 min. The concentration of TEC was adjusted to ~0.5 μM then glutaraldehyde was added to a final concentration of 0.005% (w/v). The mixture was incubated for 7 min at room temperature and the crosslinking was quenched by adding Tris-HCl, pH 8.0, to 0.1 M. The mixture was centrifuged for 10 min at 4°C, concentrated by cetrifugal filtration, and purified by gel filtration on the Superose6 INCREASE column (GE Healthcare Life Sciences) equilibrated in 20 mM Tris-HCl, pH 8.0, 150 mM KCl, 5 mM $MgCl_2$, 5 mM DTT. The eluted complex was concentrated to 3 mg/ml by centrifugal filtration (VivaProducts) before grid preparation.

### Glutaraldehyde Cross-linked Nun/TEC Preparation for CryoEM

Purified *Eco* RNAP and HK022 Nun were buffer-exchanged over the Superose 6 INCREASE column (GE Healthcare Life Sciences) equilibrated in 20 mM HEPES, pH 7.5, 150 mM KCl, 5 mM $MgCl_2$, 5 mM DTT. The TEC was assembled as before, and HK022 Nun was added at a concentration of >15 μM. Final steps of crosslinking and purification were the same as those for the TEC sample preparation.

### CryoEM grid preparation

Before freezing, 8 mM CHAPSO was added to the samples. C-flat (Protochips, Morrisville, NC) CF-1.2/1.3 400 mesh copper grids or 300 mesh gold grids were glow-charged for 15 s prior to the application of 3.5 μl of the complex sample (2.0–3.0 mg/ml protein concentration), and plunge-freezing in liquid ethane using a Vitrobot mark IV (FEI, Hillsboro, OR) with 100% chamber humidity at 22°C.

### CryoEM data acquisition and processing

The grids were imaged using a 300 keV Titan Krios (FEI) equipped with a K2 Summit direct electron detector (Gatan, Pleasanton, CA). Images were recorded with Serial EM (*Mastronarde, 2005*) in super-resolution counting mode with a super resolution pixel size of 0.65 Å and a defocus range of 0.8 to 2.6 μm. Data were collected with a dose of 8 to 10 electrons/physical pixel/s (1.3 Å pixel size

at the specimen). Images were recorded with a 15 s exposure and 0.3 s subframes (50 total frames) to give a total dose of 71 to 89 electrons/$\text{Å}^2$.

Dose fractionated subframes were 2 × 2 binned (giving a pixel size of 1.3 Å), aligned and summed using Unblur (*Grant and Grigorieff, 2015*). The contrast transfer function was estimated for each summed image using CTFFIND4 (*Rohou and Grigorieff, 2015*). From the summed images, approximately 10,000 particles were manually picked and subjected to 2D classification in RELION (*Scheres, 2012*). Projection averages of the most populated classes were used as templates for automated picking in RELION. Autopicked particles were manually inspected, then subjected to 2D classification in RELION specifying 100 classes. Poorly populated classes were removed, resulting in datasets of 470,000 particles (X-Nun/TEC), 281,000 particles (X-TEC), and 305,000 particles (TEC). These particles were individually aligned using direct-detector-align_lmbfgs software (*Rubinstein and Brubaker, 2015*). Particles were 3D autorefined in RELION using a model of *Eco* core RNAP (PDB ID 4LJZ with $\triangle 1.1\sigma^{70}$ removed; *Bae et al., 2013*) low-pass filtered to 60 Å resolution using EMAN2 (*Tang et al., 2007*) as an initial 3D template. With this initial model, 3D classifications were performed without alignment with a soft mask excluding flexible regions of the complex for more robust classification. The soft mask excluded the αII subunit, βi4, βi9, the βflap, β′i6, the β′jaw, and single-stranded nucleic acids. Among the 3D classes, the best-resolved class was 3D autorefined without the mask, and post-processed in RELION. Local resolution caculations were performed using blocres (*Cardone et al., 2013*).

## Model building and refinement

To build initial models, *Eco* core enzyme (PDB ID 4LJZ with $\triangle 1.1\sigma^{70}$ removed; *Bae et al., 2013*), DNA:RNA hybrid and downstream duplex DNA (PDB ID 2O5J; *Vassylyev et al., 2007*), and nt-strand single-stranded DNA (PDB ID 4XLN; *Bae et al., 2015*) were fit into the electron density maps using Chimera (*Pettersen et al., 2004*). These initial models were real-space refined using Phenix (*Adams et al., 2010*). In the real-space refinement, domains in the core and nucleic acids were rigid-body refined, then subsequently refined with secondary structure restraints. Single-stranded RNA and Nun were built de novo in Coot, and real-space refined with the previously refined model.

## Transcription assays

Transcription assays were done as described previously (*Vitiello et al., 2014*). Briefly, synthetic RNA was 5′-labeled with $^{32}$P-γ-ATP using T4 polynucleotide kinase (NEB) and annealed with t-strand DNA as before. His-tagged *Eco* RNAP core was added to the radio-labeled DNA:RNA hybrid, incubated for 15 min at room temperature, and nt-strand DNA was added to the mixture. After 10 min incubation, the assembled TEC was added to Ni-NTA beads in transcription buffer (20 mM Tris-HCl, pH 8.0, 40 mM KCl, 5 mM MgCl$_2$, 2 mM β-mercaptoethanol) and incubate for 20 min with shaking. Then, the immobilized TECs were washed with transcription buffer, transcription buffer containing 1 M KCl, then transcription buffer again. The TECs were then eluted with transcription buffer containing 200 mM imidazole (pH 8.0) with 0.1 mg/ml acetylated BSA. The eluted complexes were mixed with Nun (20 µM final concentration unless otherwise noted) or transcription buffer and used for transcription assays. The reactions were initiated by adding 1 mM or 0.1 mM rNTPs to the eluted TECs, and stopped by adding 2X stop buffer after 1 min or 5 min. Transcription products were resolved by 20% denaturing poly-acrylamide gel electrophoresis. The gels were exposed to phosphor screens and scanned using a Typhoon 9400 (GE Healthcare Life Sciences).

## Native mass spectrometry

The samples were buffer-exchanged into native MS buffer (500 mM ammonium acetate, 0.01% Tween-20) using Zeba microspin desalting columns (ThermoFisher Scientific) with a 40 kDa molecular weight cut-off. An aliquot (2–3 µL) of the buffer-exchanged sample was loaded into an in-house fabricated gold-coated quartz capillary and introduced into an Exactive Plus EMR instrument (ThermoFisher Scientific) using a static nanospray source. The EMR instrument was calibrated using cesium iodide. The MS parameters used included spray voltage, 1.0–1.3 kV; capillary temperature, 100°C; in-source dissociation, 10 V; S-lens RF level, 200; resolving power, 17,500 at *m/z* of 200

corresponding to 64 ms analyzer transient duration; AGC target, $5 \times 10^5$; number of microscans, 5; maximum injection time, 200 ms; high energy collision (HCD), 150 V for minimal gas-phase dissociation; injection flatapole, 8 V; interflatapole, 4 V; bent flatapole, 4 V; ultrahigh vacuum pressure, $8-9 \times 10^{-10}$ mbar; total number of scans, 100. RAW files were processed and deconvolution was performed manually using Thermo Xcalibur Qual Browser (version 3.0.63). Experimental masses were reported as average mass ± standard deviation across all the calculated mass values obtained within the charge-state distribution. The accuracies of the measured masses were calculated as [(experimental mass – theoretical mass)/theoretical mass] and ranged from 0.005% to 0.06%.

## Accession numbers

The cryoEM density maps have been deposited in the EM Data Bank with accession codes EMD-8584 (X-Nun/TEC), EMD-8585 (X-TEC), and EMD-8586 (TEC). Atomic coordinates have been deposited in the Protein Data Bank with accession codes 5UP6 (X-Nun/TEC), 5UPA (X-TEC), and 5UPC (TEC).

## Acknowledgements

We thank K Uryu and D Acehan at The Rockefeller University Electron Microscopy Resource Center for help with EM sample preparation, and M Ebrahim and J Sotiris at The Rockefeller University Cryo-EM Resource Center for help with data collection. This work was supported by a Public Health Research Institute Research Support grant (AM) and NIH grants P41 GM103314 and P41 GM109824 to BTC, R01 GM037219 to MG, and R35 GM118130 to SAD.

## Additional information

### Funding

| Funder | Grant reference number | Author |
| --- | --- | --- |
| National Institutes of Health | R35 GM118130 | Seth A Darst |
| National Institutes of Health | R01 GM037219 | Max E Gottesman |
| National Institutes of Health | P41 GM103314 | Brian T Chait |
| Public Health Research Institute Research Support grant | | Arkady Mustaev |
| National Institutes of Health | P41 GM109824 | Brian T Chait |

The funders had no role in study design, data collection and interpretation, or the decision to submit the work for publication.

### Author contributions

JYK, Conceptualization, Investigation, Methodology, Writing—original draft, Writing—review and editing; PDBO, JC, Investigation, Methodology; EAC, Conceptualization, Supervision, Methodology; AM, Conceptualization, Methodology; BTC, Methodology, Writing—original draft, Writing—review and editing; MEG, Conceptualization, Methodology, Writing—original draft, Writing—review and editing; SAD, Conceptualization, Formal analysis, Funding acquisition, Investigation, Methodology, Writing—original draft, Writing—review and editing

### Author ORCIDs

Seth A Darst, http://orcid.org/0000-0002-8241-3153

## Additional files

### Supplementary files

• Supplementary file 1. Model statistics from Molprobity (Chen et al., 2010).

• Supplementary file 2. Results of superpositions of TEC models.

## Major datasets

The following datasets were generated:

| Author(s) | Year | Dataset title | Dataset URL | Database, license, and accessibility information |
|---|---|---|---|---|
| Kang JY, Darst SA | 2017 | CryoEM structure of HK022 Nun - E. coli RNA polymerase elongation complex | http://www.rcsb.org/pdb/explore/explore.do?structureId=5UP6 | Publicly available at the RCSB Protein Data Bank (accession no: 5UP6) |
| Kang JY , Darst SA | 2017 | CryoEM structure of HK022 Nun - E. coli RNA polymerase elongation complex | https://www.ebi.ac.uk/pdbe/entry/emdb/EMD-8584 | Publicly available at the EMBL-EBI Protein Data Bank in Europe (accession no: EMD-8584) |
| Kang J, Darst SA | 2017 | CryoEM structure of crosslinked E. coli RNA polymerase elongation complex | http://www.rcsb.org/pdb/explore/explore.do?structureId=5UPA | Publicly available at the RCSB Protein Data Bank (accession no: 5UPA) |
| Kang JY, Darst SA | 2017 | CryoEM structure of crosslinked E. coli RNA polymerase elongation complex | https://www.ebi.ac.uk/pdbe/entry/emdb/EMD-8585 | Publicly available at the EMBL-EBI Protein Data Bank in Europe (accession no: EMD-8585) |
| Kang JY, Darst SA | 2017 | CryoEM structure of E.coli RNA polymerase elongation complex | http://www.rcsb.org/pdb/explore/explore.do?structureId=5UPC | Publicly available at the RCSB Protein Data Bank (accession no: 5UPC) |
| Kang JY, Darst SA | 2017 | CryoEM structure of E.coli RNA polymerase elongation complex | https://www.ebi.ac.uk/pdbe/entry/emdb/EMD-8586 | Publicly available at the EMBL-EBI Protein Data Bank in Europe (accession no: EMD-8586) |

The following previously published datasets were used:

| Author(s) | Year | Dataset title | Dataset URL | Database, license, and accessibility information |
|---|---|---|---|---|
| Darst SA, Bae B | 2013 | Crystal Structure Analysis of the E. coli holoenzyme | http://www.rcsb.org/pdb/explore/explore.do?structureId=4LJZ | Publicly available at the RCSB Protein Data Bank (accession no: 4LJZ) |
| Vassylyev DG, Vassylyeva MN | 2007 | Crystal structure of the T. thermophilus RNAP polymerase elongation complex with the NTP substrate analog | http://www.rcsb.org/pdb/explore/explore.do?structureId=2O5J | Publicly available at the RCSB Protein Data Bank (accession no: 2O5J) |

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
