## [Decision Letter]

Thank you for submitting your article "Structural Basis of Transcription Arrest by Coliphage HK022 Nun in an *Escherichia coli* RNA Polymerase Elongation Complex" for consideration by *eLife*. Your article has been favorably evaluated by Gisela Storz (Senior Editor) and three reviewers, one of whom, Nikolaus Grigorieff (Reviewer #1), is a member of our Board of Reviewing Editors. The following individual involved in review of your submission has agreed to reveal their identity:; Patrick Cramer (Reviewer #2).

The reviewers have discussed the reviews with one another and the Reviewing Editor has drafted this decision to help you prepare a revised submission.

Summary:

The authors present high-resolution cryo-EM structures of bacterial RNA polymerase ternary elongation complexes (TECs) with and without the translocation-blocking factor Nun. In the Nun-TEC structure, the functionally important C-terminal 23 residues of Nun could be observed. The authors interpret previously existing data and provide new biochemical experiments in light of the observed Nun-nucleic acid and Nun-RNAP interactions. The Nun-TEC structure along with in vitro functional analysis of Nun mutants support a mechanism of translocation inhibition due to the formation of numerous Nun interactions with RNAP and nucleic acids, which would disfavor sliding of nucleic acids with respect to RNAP. For Nun to bind, the polymerase cleft needs to open slightly, consistent with a function of Nun during transcriptional pausing. This work establishes the molecular mechanism of Nun protein function in bacterial transcription.

Essential revisions:

We recommend publication of this excellent manuscript in *eLife* after the following questions/comments have been addressed/discussed:

1) Was glutaraldehyde cross-linking necessary to stabilize the TEC-20A construct (sample instability was only observed for the TEC-9A sample)?

2) The authors mention in the Introduction that Nun function is dependent on NusA, NusB, NusE, and NusG. Do the new structures shed light on the mechanistic basis of this dependence?

3) Is it possible that the cross-linking may have "created" some of the extensive contacts seen between Nun and the rest of the complex?

4) Can the authors comment on the role of Nun's N-terminus in Nun loading, transcription termination, and prevention of non-specific interactions with non-λ DNA?

5) Nun Trp108 was found to be important for transcriptional arrest and termination, possibly by intercalating into downstream DNA (Watnick & Gottesman, Science 1999; Watnick et al., Genes Dev. 2000). An Ala or Leu substitution of Trp108 abrogated function in vivo (to <2% of wild-type), yet the authors report only a modest decrease in function in vitro (~66% relative to wild-type, Figure 5). There is additional evidence for contacts with DNA of Trp108 from crosslinking studies, which showed that a W108A variant did not crosslink to template, while wild-type did (Watnick & Gottesman, Science 1999). However, the authors state "this [binding to DNA] was not observed in the structure." The authors should explain how their results could be reconciled with these previous reports.

6) Related to the previous point, could the authors elaborate on a possible mechanism by which regulators whose binding and RNAP modulation activity are uncoupled (e.g. the Nun W108A and R94A) might work in this system?

7) Some of the other in vitro data presented by the authors seem to be at odds with previously published in vivo data: a) H98A was shown to abolish termination activity (Watnick et al., Genes Dev. 2000); in the present study, this mutant retains 60% wild-type activity. b) W108A shows no in vivo activity (Watnick et al., Genes Dev. 2000); in the present study, the activity is 66% of wild-type. The authors should discuss these apparent discrepancies.

8) The local resolution estimates generated by ResMap are very unreliable and can only be used to establish relative resolution differences between regions of the density, not absolute values (see for example, Zubcevic et al., Nat. Struct. Mol. Biol., 2016). The authors should provide other evidence to support their resolution claim or rephrase the relevant sections in the manuscript.

---

## [Author Response]

*Essential revisions:*

*We recommend publication of this excellent manuscript in eLife after the following questions/comments have been addressed/discussed:*

*1) Was glutaraldehyde cross-linking necessary to stabilize the TEC-20A construct (sample instability was only observed for the TEC-9A sample)?*

We did not test the stability of the Nun/TEC-20A sample without glutaraldehyde crosslinking. Nun only interacts with the RNA transcript between -8 to -6 (Figure 4), so we would not expect that extending the RNA upstream from -9 would have any effect on Nun stability.

*2) The authors mention in the Introduction that Nun function is dependent on NusA, NusB, NusE, and NusG. Do the new structures shed light on the mechanistic basis of this dependence?*

The mechanism of Nus factor stimulation of Nun activity is unknown, but is specific for λ transcripts containing boxA of the *nut* sequence. The lack of structural information for any of the Nus factors in complex with RNAP precludes speculation. Results are consistent with the Nus factors simply stabilizing the binding of Nun to λ *nut* TECs. A new paragraph is included in the Discussion about this issue (subsection “Mechanism of Nun translocation arrest”, second paragraph) as well as a new schematic diagram (Figure 7).

*3) Is it possible that the cross-linking may have "created" some of the extensive contacts seen between Nun and the rest of the complex?*

The level of glutaraldehyde crosslinking is very low. Note the presence of apparently stoichiometric non-crosslinked Nun evident in the post gel-filtration sample (Figure 1—figure supplement 2), indicating that most, if not all of the Nun molecules were crosslink free. Thus, it is very unlikely that the crosslinking introduced non-native contacts/conformations. The stabilization of Nun binding in the complex does not come from direct Nun-RNAP crosslinking, but from intramolecular RNAP-RNAP crosslinks, which dominate the sample. This has been clarified (subsection “Structure determination of the Eco TEC ± Nun”, second paragraph).

The correspondence of the X-TEC20A and TEC20A structures also supports this conclusion (Figure 1—figure supplement 5), as already pointed out in the text (–subsection “Structure determination of the Eco TEC ± Nun”, second paragraph).

*4) Can the authors comment on the role of Nun's N-terminus in Nun loading, transcription termination, and prevention of non-specific interactions with non-λ DNA?*

Discussion added (subsection “Mechanism of Nun translocation arrest”, second paragraph) and added schematic Figure 7.

*5) Nun Trp108 was found to be important for transcriptional arrest and termination, possibly by intercalating into downstream DNA (Watnick & Gottesman, Science 1999; Watnick et al., Genes Dev. 2000). An Ala or Leu substitution of Trp108 abrogated function in vivo (to <2% of wild-type), yet the authors report only a modest decrease in function in vitro (~66% relative to wild-type, Figure 5). There is additional evidence for contacts with DNA of Trp108 from crosslinking studies, which showed that a W108A variant did not crosslink to template, while wild-type did (Watnick & Gottesman, Science 1999). However, the authors state "this [binding to DNA] was not observed in the structure." The authors should explain how their results could be reconciled with these previous reports.*

In Watnick et al. (1999), wt-Nun was extended at the C-terminus by a Cys residue (so Cys110) and shown to have wt activity. The new C-terminal Cys residue was then modified with a crosslinking reagent having a 15 Å linker arm length. This modified Nun was shown to crosslink to downstream duplex DNA. In the X-Nun/TEC20A structure, the normal C-terminal Cα of Nun (S109) is already only 13 Å from the downstream duplex DNA, so extending the C-terminus by another residue (C110) and then a crosslinker with a 15 Å linker arm could easily reach the DNA as observed. We believe the lack of crosslinking for the W108 substitutions can be explained by lack of binding of the Nun mutants at the low concentrations of the reactions.

We believe the discussion already present (subsection “Nun interacts with RNAP, DNA, and RNA”, fourth paragraph):

“β’D329 forms part of a pocket in the RNAP structure that accommodates Nun W108 (Figure 4). The structure is consistent with previous findings that the C-terminus of Nun is within 15-20 Å of the downstream duplex DNA (Watnick and Gottesman, 1999). The location of the Nun C-terminus within the vicinity of the downstream duplex DNA, combined with the finding that normal Nun activity required an aromatic residue at Nun position 108 led to the proposal that Nun-mediated translocation arrest was facilitated by the intercalation of Nun W108 into the downstream DNA (Watnick and Gottesman, 1999) but this is not observed in the structure.”

combined with the new discussion (subsection “Mechanism of Nun translocation arrest”, last paragraph):

‘Nun mutants H98A, K106A/K107A, and W108A have been shown previously to be significantly (K106A/K107A) or totally (H98A, W108A) defective in Nun function both in vivoand in vitro{Watnick:1999ve, Watnick:2000wm, Kim:2004hw}. […] Thus, the previous studies assessed mutant Nun activity at [Nun] too low to overcome the binding defects introduced by the mutations.’

clarifies this point.

*6) Related to the previous point, could the authors elaborate on a possible mechanism by which regulators whose binding and RNAP modulation activity are uncoupled (e.g. the Nun W108A and R94A) might work in this system?*

We don’t completely understand this question. The point of this discussion (starting on the first full para. of pg. 13) is that for many regulatory factors, RNAP binding and regulatory function are not coupled (so, for example, single amino acid substitutions of Gre factors eliminate their endonucleolytic transcript cleavage activity but do not have a noticeable effect on RNAP binding). Nun appears to fall into a different category, where binding and activity are essentially one and the same. Although binding and activity of Nun mutants are not strictly correlated (Figure 5), each of the tested mutants show significant activity when assayed at saturating concentration despite widely varying effects on RNAP binding.

*7) Some of the other in vitro data presented by the authors seem to be at odds with previously published in vivo data: a) H98A was shown to abolish termination activity (Watnick et al., Genes Dev. 2000); in the present study, this mutant retains 60% wild-type activity. b) W108A shows no in vivo activity (Watnick et al., Genes Dev. 2000); in the present study, the activity is 66% of wild-type. The authors should discuss these apparent discrepancies.*

Our results are actually not discrepant with previous findings. It is true that H98A and W108A were previously found to have no activity in vivoand in vitro, but these assays were done at very low levels of Nun (200 nM or less), where the binding defects introduced by the mutations would prevent binding and thus function (see Figure 5). Our assays demonstrating significant function for these mutants were done at saturating concentrations of Nun (> 250 µM) to overcome the binding defects. This point is clarified in a new paragraph of the Discussion (subsection “Mechanism of Nun translocation arrest”, last paragraph).

*8) The local resolution estimates generated by ResMap are very unreliable and can only be used to establish relative resolution differences between regions of the density, not absolute values (see for example, Zubcevic et al., Nat. Struct. Mol. Biol., 2016). The authors should provide other evidence to support their resolution claim or rephrase the relevant sections in the manuscript.*

We became aware of this issue after submitting our manuscript. We have redone the local resolution calculations using blocres (Cardone et al., 2013, J. Struct. Biol.) and updated the Figures (Figure 1—figure supplement 6, Figure 1—figure supplement 7).